# Polydopamine Copolymers for Stable Drug Nanoprecipitation

**DOI:** 10.3390/ijms232012420

**Published:** 2022-10-17

**Authors:** Danna Niezni, Yuval Harris, Hagit Sason, Maytal Avrashami, Yosi Shamay

**Affiliations:** Faculty of Biomedical Engineering, Technion–Israel Institute of Technology, Haifa 3200003, Israel

**Keywords:** polydopamine, nanoparticles, nanoprecipitation, drug delivery, nanomedicine, indolium, colon cancer, aggregation-induced emission

## Abstract

Polydopamine (PDA), a biomaterial inspired by marine mussels, has attracted interest in cancer nanomedicine due to its photothermal properties, nanoparticle coating, and pi-pi stacking-based drug encapsulation abilities. Despite numerous one-pot and post-polymerization modifications, PDA copolymers have not been sufficiently studied in the context of stabilizing hydrophobic drugs in the process of nanoprecipitation. In this study, we tested combinatorial panels of comonomers with PDA to optimize drug loading efficiency, particle size and stability of nano formulations made via drug nanoprecipitation. As a selection criterion for optimal comonomers, we used drug aggregation-induced emission (AIE). We identified 1,1,2-Trimethyl-3-(4-sulfobutyl)benz[e]indolium (In820) as a novel and highly useful comonomer for catecholamines and optimized the conditions for its incorporation into PDA copolymers used for drug nanoprecipitation. Surprisingly, it was superior to polyethylene glycol modifications in every aspect. The leading copolymer, poly(dopamine)-poly(L-dopa)-co-In820 (PDA-PDO-In820 1:1:1), was shown to be a good stabilizer for several hydrophobic drugs. The resulting nanoparticles showed stability for up to 15 days, high encapsulation efficiency of at least 80%, low toxicity, and high antitumor efficacy in vitro. Nanoprecipitation of hydrophobic drugs can be greatly enhanced by the use of PDA copolymers containing In820, which are easy-to-prepare and highly effective stabilizers.

## 1. Introduction

Polydopamine (PDA) is highly popular in biomedical research due to its simple preparation, good biocompatibility, and versatility in bioconjugation [1,2]. Although the specific synthetic mechanism and absolute structure of PDA auto-oxidative polymerization in alkaline buffers have not been clarified, it has been deemed useful in many fields, such as energy, environment, and biomedicine [3]. In biomedicine, specifically, it has attracted significant interest in the field of tumor-targeted drug delivery due to its photothermal abilities, nanoparticle coating, and drug encapsulation via pi-pi stacking [4,5].

Compared to other drug delivery systems, PDA has several advantages. First, the preparation conditions of PDA are simple and mild [2]. No organic solvent is required for PDA synthesis, providing great convenience for its research and application. In addition, due to the superior adhesion of PDA, it is also possible to coat various types of organic and inorganic nanoparticles [6]. Furthermore, the catechol and anthracene fractions found on PDA surface allow secondary modification with thiol or amino group-constraining compounds through the Michael addition reaction or Schiff base reaction under alkaline conditions [1,7].

Though as a homopolymer it is incredibly useful for coating existing materials, it is rarely used alone for systemic administration of drug delivery systems. The most common post-polymerization modification for PDA-based systems is PEGylation [8,9], which has been used, for example, for delivery of doxorubicin and hydroxycamptothecin with high encapsulation efficiency [10]. 

Rather than post-polymer modifications, one-step methods have been recently proposed for the preparation of PDA copolymers with comonomers containing catechol and acrylate or amine groups. An interesting example is polydopamine–polyethyleneimine (PDA-PEI) copolymer which has intrinsic fluorescence orders of magnitude more efficient than PDA and melanin [11]. However, PDA copolymers have not been studied in the context of drug nanoprecipitation. In this method, the drug is dissolved in an organic solvent and added gradually to an aqueous solution [12,13]. The nanoparticles will form spontaneously in the presence of a stabilizer, resulting in a high encapsulation ratio of the drug in the nanoparticles [14]. Most works propose PDA shells that are loaded with drug instead of the co-precipitation method [10]. Since micelles and liposomes usually have a low encapsulation ratio relative to nanoprecipitation [13], we hypothesized that incorporating the advantages of dopamine copolymers with nanoprecipitation can improve the ability to stabilize nanoparticles for many drugs.

We used drug aggregation-induced emission (AIE) to analyze the overall stability of nanoparticles. AIE refers to increased fluorescence emission in the aggregated state over the soluble state [14,15,16,17]. If an AIE drug is fluorescent, we assume the nanoparticles were not stable and therefore it aggregated [14].

In this study, we screened for comonomers using AIE and identified a new hybrid of catecholamines with 1,1,2-Trimethyl-3-(4-sulfobutyl)benz[e]indolium (In820) [14] and the conditions for its formation. This material is poly(dopamine)-poly(L-dopa)-co-In820 (PDA-PDO-In820). We characterized the resulting copolymer and nanoparticles stabilized by it, as well as tested their efficacy in vitro. The particles are stable, have high drug loading, and are at least as good as free drugs in 2D and 3D models.

## 2. Results and Discussion

### 2.1. Finding the Optimal Conditions for Polymerization

Six catecholamines and related aromatic amine monomers were tested in five different conditions and DDW as control (Figure 1a). Besides dopamine, we tested L-dopa, norepinephrine, serotonin, tyramine, and tryptamine. All monomers were at a final concentration of 2 mg/mL, as described in previous studies [18]. Tris and sodium bicarbonate buffers (pH 9) are well-known alkaline buffers for polymerization of catecholamines. They were compared to a NaOH solution at the same pH level and a hydrogen peroxide solution to test the effect of oxidation. In order to evaluate the progression of the reactions, the absorbance of the solutions was measured over 4.5 h. As expected, the most dominant change was in Tris and bicarbonate buffers [4] (Figure 1b,c, left panels). The process, observed by the change in color, was significantly faster in the sodium bicarbonate buffer than in tris buffer. Therefore, we chose this buffer for the copolymerization process as well. We measured the emission spectra of dopamine (ex. at 400 nm) and L-dopa (ex. at 420 nm) and found that they match the expected peaks of catecholamines after polymerization [19] (Figure 1b,c, middle panels). ATR analysis of poly(dopamine) and poly(L-dopa) (PDO) in comparison to the monomers (Figure 1b,c, right panels) revealed peaks at 2936 and 3335 cm^−1^, which correlate to the primary amine in dopamine. The decrease in the transmittance after the polymerization indicates the expected ring closing. The change in the transmittance at 2400–3500 cm^−1^ hints at the disappearance of the carboxylic acid in L-dopa. It was suggested that part of its oxidation process involves losing the carboxylic acid [20].

The change over time in the absorbance spectra of the different monomers in sodium bicarbonate buffer is shown in Figure 1d. Dopamine and L-dopa polymerization process and change of color are faster and more apparent than norepinephrine and serotonin, while tyramine and tryptamine did not change color at all. The absorbance spectra of PDA and PDO correlate to the expected spectra for poly(catecholamine)s [21]. We hypothesize that the ability to polymerize under these conditions relates to the monomers’ structure. Dopamine, L-dopa, and norepinephrine are all catecholamines and have two hydroxyl groups on the benzene ring, while serotonin and tyramine have only one hydroxyl group, and tryptamine has no OH groups. The polymerization process is not fully understood but is known to involve the oxidation of the monomers and formation of a quinone, which can only happen with two hydroxyls [2,22].

### 2.2. Optimal Copolymer for Hydrophobic Drugs Stabilization

Poly(catcholamine)s are hydrophobic polymers and will not be able to stabilize hydrophobic drugs because they precipitate on their own in water. Therefore, we sought a comonomer that would prevent the precipitation of the copolymer, as well as stabilizes nanoparticles of hydrophobic drugs prepared by nanoprecipitation. To find the optimal monomer combination, a panel of comonomers was tested. The panel included known PDA comonomers and modifiers such as PEG-SH [23], PEG-methacrylate [24,25], and HPMA [26]. In addition, we tested sulfated indolium molecules, which are known to react with aldehydes in fluorescent dye synthesis [27]. We hypothesized that if PDA can react with thiols, amines, and methacrylates it might be able to react with indoliums as well. 

We first tested the copolymer’s ability to stabilize nilotinib, a hydrophobic kinase inhibitor, by utilizing its aggregation-induced emission (AIE) properties. Based on previous work in our lab, we precipitated nilotinib with bicarbonate buffer and the different copolymers, in a 96-well plate and measured the relative surface area (confluence) that is blue under DAPI filter (ex. at 377 nm, em. at 477 nm), where higher confluence correlates with larger particle size. An illustration of the colors formed after 4 h reaction is shown in Figure 2a (left panel). Some of the wells turned brown due to the catecholamines polymerization reaction, while wells containing In820 and In783 turned purple and red, respectively, since they are precursors for dyes. In780 is also a dye precursor, its reaction in the bicarbonate buffer with and without the catecholamines formed different colors. Next, we added nilotinib to all the wells, and the aggregation confluence was normalized to nilotinib aggregates in sodium bicarbonate buffer (0.1 M) and shown in Figure 2a (right panel). The lowest intensity measured was in copolymers containing In820 and SDS, indicating their superiority over PEG and HPMA. However, SDS containing copolymers resulted in aggregation after a nanoparticle purification process, thus we decided to further study In820 copolymers with different catecholamines and not SDS. 

We further validated these results and measured the ability of different PDA copolymers to stabilize sorafenib and nilotinib with dynamic light scattering (Figure 2b,c). Sorafenib, like nilotinib, is a hydrophobic multi-kinase inhibitor used extensively in nanoparticles made via nanoprecipitation. According to previous studies, sorafenib is more likely to form stable nanoparticles than nilotinib [13]. As mentioned above, the homopolymers of dopamine, L-dopa and norepinephrine were indeed unable to form stable nanoparticles with sorafenib and nilotinib and precipitated on their own. Surprisingly, PEGylation of the polymers did not succeed in stabilizing sorafenib or nilotinib. The only copolymers which stabilized both drugs in nanoparticles smaller than 150 nm and with polydispersity index of around 0.25 were those containing In820 (Figure 2b,c). They were also the only copolymers that were completely soluble in water. We focused on combinations without norepinephrine due to the slow polymerization kinetics. To further evaluate the comonomers, we measured the stability of nilotinib nanoparticles over long time periods stabilized by either IR783, PDO-In820, PDA-In820, or PDA-PDO-In820 (Figure 2d). We found that nanoparticles stabilized with IR783, aggregated within the first 24 h while nanoparticles stabilized by the In820 containing copolymers remained stable for longer than two weeks. The optimal formulation with the smallest size and highest encapsulation ratio was PDA-PDO-In820. The encapsulation ratios with PDA-In820 and PDO-In820 were fairly low only 45% and 12% (respectively) compared to 89% for the PDA-PDO-In820 nanoparticles, and their size remained lower than 150 nm after 17 days. 

Then, we compared the three leading copolymers with nine different hydrophobic drugs. We tested a panel of drug formulations with PDA-PDO-In820, PDA-In820, or PDO-In820 and monitored their stability over three days (Figure 3). Drugs that were stable for two days were measured on the third day as well. Nilotinib and cyclosporine are the only drugs that were successfully stabilized by all three polymers for three days. PDO-In820 was the least successful among the three of them. It was only able to stabilize nilotinib and cyclosporine for three days, while irinotecan did not form nanoparticles at all. The difference in the stabilization ability between PDA-In820 and PDA-PDO-In820 was evident with trametinib. The nanoparticles aggregated after two days when stabilized with PDA-In820 but remained stable for three days when stabilized by PDA-PDO-In820.

The final comparison was based on the morphology of the nanoparticles as captured by SEM imaging (Figure 4). For this, we chose sorafenib as the model drug. With all three stabilizers, the particles were spherical. Some aggregates were seen, probably due to the sample preparation and drying processes. The sizes measured in the SEM images were close to the expected sizes: 61 ± 16 nm for PDA-PDO-In820, 64 ± 17 nm for PDA-In820, and 67 ± 22 nm for PDO-In820. The differences between the DLS and SEM measurements were less than 25%. Based on these images, we conclude that the morphology and size of the three formulations are the same.

### 2.3. Optimization of PDA-PDO-In820 Synthesis

After we established that PDA-PDO-In820 can stabilize the largest variety of drugs, forms stable nanoparticles for the longest periods of time, and has the highest encapsulation efficiency, we sought to further characterize it and optimize its synthesis. Since the structures of polydopamine and poly(L-dopa) are unknown, we did not aspire to identify the exact structure of the copolymer. 

Different ratios of the three monomers were tested and showed similar or inferior stabilizing abilities. The 1:1:1 ratio was chosen due to superior stabilization ability and reproducibility. We tested both the size and relative drug encapsulation in the nanoparticles over time from the beginning of the polymerization process. We noticed that the polymer requires an aging period of a week before it can form stable nanoparticles with high drug loading. The drug loading calculation was done for nilotinib nanoparticles (Figure 5a,b). In the first hour and a half of the polymerization process, its precipitation capability was not efficient. Even though the polymer solution had dark color after the first 15 min, the drug-containing solution was still pale, with a total drug loading of ~10%. After 24 h, the nanoparticle suspension had a pale brown color with a drug loading of 55%. Only after seven days, we saw a significant increase in both the drug loading (78%) and the color of the suspension. 

We also measured the size and stability of ponatinib nanoparticles with different reaction times (Figure 5c). When formulating nanoparticles of ponatinib with freshly made copolymer (up to 5 h from polymerization initiation), microparticles are formed instead of nanoparticles. Particles stabilized with PDA-PDO-In820 5 h and 24 h from the polymerization initiation were in the desired size range of below 150 nm. 

During this process, we also found that different drugs form better particles by different purification methods. Nilotinib nanoparticles purified in centrifugation were not stable for more than a day, while their stability was maintained for 2 weeks when purified in a Sephadegx G25 PD10 desalting column. On the other hand, ponatinib only forms stable nanoparticles when purified in centrifugation. We performed a cumulative drug release profile for ponatinib nanoparticles stabilized with PDA-PDO-In820 and found a relatively linear profile with 80% release within 20 h (Figure 5d).

### 2.4. In Vitro Toxicity and Antitumor Efficacy Experiments

In order to evaluate the safety of our newly formulated polymers, we tested in vitro cytotoxicity of the three leading copolymers using mouse embryonic fibroblast cells (3T3 cells) as a non-cancerous model (Figure 6). The copolymers were purified with Sephadex G25 columns and were serially diluted in a 96-well plate seeded with the cells. We used two separate viability assays: MTT and an image-based cell counting method using automated microscopy. In the MTT assay, we found no reduction in cell viability over control, even in the highest concentration of polymers (Figure 6a). Similar results were evident when we performed cell counting under the same experimental conditions (Figure 6b). As the image-based cell counting is a non-invasive procedure, we followed cell proliferation over time and found that in agreement with the MTT assay, the copolymers are not only non-toxic, but they are also non cyto-static and allow cell proliferation, even in the highest concentration (Figure 6c–e). The cells’ morphology was not changed following incubation with the polymers, though at higher concentrations, intracellular dark vesicles could be observed, indicating cellular uptake through the endo-lysosomal pathway [28] (Figure 7). 

To further demonstrate the non-toxic effect of the copolymers, enzalutamide nanoparticles were formulated and tested as non-toxic particles. It is a selective competitive androgen receptor inhibitor used for prostate cancer which is not expected to have a toxic impact on fibroblast cells that are not dependent on this receptor [29,30]. Free enzalutamide had slight toxicity in high concentrations, which was attributed to the >5% DMSO in the medium (Figure 8a). In high concentrations (0.1 and 0.01 mg/mL) of the nanoparticles, growth inhibition and cell death were observed (Figure 8b–e). Similarly to the images in Figure 7, intracellular dark vesicles were seen with higher concentrations and might have caused the observed cell death (Figure 9). 

Antitumor efficacy experiments were done on HCT116 cell line. Since the main driving mutation in this cell line is in the Ras proto-oncogene, it is known to be sensitive to trametinib [31]. Sorafenib, a multi-kinase inhibitor, was also tested as a less efficacious drug [32]. Both drugs were tested as free drugs and as nanoparticles stabilized by PDA-PDO-In820 on 2D and 3D in vitro models.

Both formulations were imaged in HR-SEM (Figure 10a) and demonstrated that the nanoparticles are spherical. However, some aggregates are visible, which might be accounted for by the sample preparation technique for the imaging. The average trametinib nanoparticles size, according to the SEM images, is 68 nm, and, according to DLS measurements, is 130 ± 7 nm. While for sorafenib nanoparticles, the DLS and SEM measurements are more in agreement 74 and 61 nm, respectively. Since particles are expected to shrink in size during the SEM sample preparation process, these results are logical.

The results shown in Figure 10b,c are normalized to the encapsulation efficiency of the nanoparticles, which according to the UHPLC measurements, is 90% for trametinib and approximately 80% for sorafenib. In 2D as well as in the 3D cell cultures, the trend is very clear, trametinib is much more efficacious in killing this cell line (*p* = 0.0125, 0.0024 respectively). Moreover, this trend and the drugs’ efficacy are maintained when encapsulating them in nanoparticles. After several days of incubation with the treatment, we noticed differences in the morphology of the spheroids that derive from their viability. The spheroids grew in the non-treated wells or wells where the treatment was not effective enough but shrunk or maintained their size in other wells (Figure 10d,e). The differences between the high concentrations of the drugs in both free drugs and nanoparticles are more significant in trametinib than sorafenib, as expected. In high concentrations, the effect and change in size is significant in both trametinib (*p* < 0.005) and sorafenib (*p* < 0.05) treatments. While the difference in size between the control and the low sorafenib concentration is non-significant, the difference between it and low trametinib concentration is (*p* < 0.05).

To conclude, in this work, we have discovered PDA-PDO-In820, which is highly useful for stabilization and encapsulation of hydrophobic drugs. It was shown to be non-toxic and safe in fibroblasts in vitro, and efficacious in delivering drugs to colon cancer cells. The formulations present a similar antitumor impact as the free drug trametinib both in 2D and 3D in vitro models without free polymer toxicity.

## 3. Methods

### 3.1. Materials and Reagents

All non-drug chemicals were purchased from Sigma Aldrich (St. Louis, MO, USA). DMSO was purchased from Carlo Erba (Emmendingen, Germany). All drugs were purchased from LC-Laboratories (Woburn, MA, USA) and MedChemExpress (Monmouth Junction, NJ, USA).

### 3.2. Polymerization Process

Conditions matrix: Dopamine (Sigma Aldrich), L-dopa (Sigma Aldrich), norepinephrine (Sigma Aldrich), serotonin (Holland Moran), tyramine (Sigma-Aldrich), and tryptamine (Holland Moran, Yehud, Israel) were dissolved in DDW to a concentration of 4 mg/mL. Polymerization conditions concentrations: Tris(hydroxymethyl)aminomethane (tris buffer, Sigma-Aldrich) 0.005 M, sodium bicarbonate buffer (Biolab, Jerusalem, Israel) 0.1 M, NaOH (Biolab) 10^−5^ M, hydrogen peroxide (H_2_O_2_, Merck, Kenilworth, NJ, USA) 15% in DDW. 

Comonomers matrix: The reactions were in sodium bicarbonate buffer 0.1 M. Polyethylene glycol-SH (PEG-thiol, Mn = 1 kDa, Creative PEGWorks, Chapel Hill, NC, USA), 1,1,2-Trimethyl-3-(4-sulfobutyl)benz[e]indolium, inner salt (In820, Sigma-Aldrich), 2,3,3-Trimethyl-1-(4-sulfobutyl) indolium (In783, Sigma-Aldrich). 1,2,3,3-Tetramethyl-3H-indol-1-ium iodide (In780, Holland Moran), N-(2-Hydroxypropyl) methacrylamide (HPMA, Sigma-Aldrich), Sodium Lauryl Sulfate (SDS, Spectrum, New Brunswick, NJ, USA) were dissolved to a concentration of 2 mg/mL and Polyethylene glycol methyl ether methacrylate (PEG-MA, Mn = 2 kDa, Sigma-Aldrich) in a final concentration of 7.5 mg/mL. Two hundred microliters of the monomer-comonomer (50:50% v) solution were added to wells in a 96-well plate and incubated covered at room temperature for 4 h.

Polymerization of copolymers: For the dual copolymers, 4 mg of each monomer were dissolved in 1 mL of sodium bicarbonate buffer (0.1 M) and mixed, final concentration of each monomer was 2 mg/mL. For the triple copolymers, 6 mg of each monomer were dissolved in 1 mL of sodium bicarbonate buffer (0.1 M) and mixed, final concentration of each monomer 2 mg/mL. The copolymers were shaken at 800 rpm for 4 h at room temperature, stored in a dark place, and aged for 7 days before used.

### 3.3. Absorbance Measurements

Absorbance and emission spectra were evaluated with Synergy H1 (BioTek^®^_,_ Santa Clara, CA, USA) plate reader. Absorbance was measured every 30 min and emission spectra were measured 4.5 h from the beginning of the polymerization process, each polymer was excited according to its absorbance spectra.

### 3.4. ATR-FTIR

Samples were lyophilized by Freeze zone 2.5 (Labconco, Kansas City, MO, USA) at −54 °C, 0.01 mbar for 48 h. Transmittance was measured in Bruker’s (Billerica, MA, USA) Fourier transform infrared (FT-IR) Alpha at 400–4000 cm^−1^ to identify the chemical signature of the polymers. The measurements were done at the Chemical and Surface Analysis Lab at The Schulich Faculty of Chemistry, the Technion.

### 3.5. Imaging Fluorescent Drug Aggregates with Automated Microscopy 

To image fluorescent drug aggregates, 10 µL of nilotinib was added (1 mg/mL) to 60 µL bicarbonate buffer (0.1 M) and 20 µL of the copolymers to wells in a 96-well plate. The wells were imaged in the DAPI channel (Ex 377 nm Em 447 nm) of LionHeart (BioTek^®^) with a 10 ms exposure 10% digital gain, 100% LED intensity, and image-based autofocus. AIE effect quantification was done by calculating the relative confluence of blue emission in each well compared to nilotinib aggregates in water. 

### 3.6. Preparation of Nanoparticles

Drugs dissolved in DMSO (10 mg/mL) were added under slight vortex to aqueous dye/copolymer solution, buffered with 0.1M sodium bicarbonate. IR783 (Sigma-Aldrich) and the polymers’ concentrations are 2 mg/mL. The solution was either centrifuged (30,000× *g*, 15 min, R.T.) and the pellet was resuspended in 1ml of DDW or purified using a PD-10 desalting column (GE Healthcare, Chicago, IL, USA) and 1.7 mL of eluent was collected. If the particles were cleaned in centrifuge, the pellet resuspension was sonicated using Sonics’ (Newtown, CT, USA) Vibra-cell ultrasonic processor (20% amplitude, 3 s pulses) until homogenous.

### 3.7. Characterization of Nanoparticles

Size: Dynamic light scattering (DLS) measurements were conducted using a Zetasizer Nano ZS (Malvern Panalytical, Malvern, UK). 

Drug loading: Nanoparticles diluted 1:10 in acetonitrile:ethanol (50:50) solution were measured with ultra-performance liquid chromatography (Acquity arc UHPLC, Waters Corp, Milford, MA, USA) using a CORTECS C18 (4.6 × 50 mm 2.7 µm) column. 

### 3.8. HR-SEM

Two microliters of the nanoparticle solution were applied to a silicon wafer and placed in a desiccator under vacuum for 72 h for dehydration. High-resolution scanning electron microscopy (HR-SEM) imaging was performed by the Technion EMC, the Electron Microscopy Center on a Zeiss Ultra Plus high-resolution SEM (Oberkochen, Germany), equipped with a Schottky field-emission gun. Specimens were imaged at acceleration voltages of 1.3 kV and a working distance of 2.4 mm. The Everhart Thornley (“SE2”) secondary electron imaging detector was used.

### 3.9. Cell Cultures

3T3 cells were provided by the Heller lab at MSKCC, and HCT 116 cells were a kind gift provided by the lab of Moshe Elkabets, Ben-Gurion University, Israel. The cells were cultured in Dulbecco’s Modified Eagle Medium (DMEM, Sartorius, Goettingen, Germany) supplemented with 10% FCS, 2 mM L-glutamine, Penicillin G Sodium Salt: 100 units/mL and Streptomycin Sulfate: 0.1 mg/mL (pen-strep). All cells were incubated at 37 °C with 5% CO_2_ and 65% humidity.

### 3.10. Polymers Toxicity Assay in 2D 

3T3 cells were seeded in 96-well plates at a 30% confluency (6000 cells per well) and allowed to adhere for 24 h. Polymers were purified with Sephadex G25 columns (PD10 desalting column), nanoparticles were suspended in DDW while the free drug was dissolved in DMSO. Untreated cells (control) were used to establish 100% viability. The effect on cell viability was measured with MTT (5 mg/mL, Glentham Life Sciences, Corsham, UK).

### 3.11. Cell Viability Assay in 2D 

HCT 116 cells were seeded in 96-well plates at a 30% confluency (5000 cells per well) and allowed to adhere for 48 h. All drugs were dissolved in DMSO, nanoparticles were suspended in DDW, and added to the wells. Untreated cells (control) were used to establish 100% viability. The effect on cell viability was measured with Promega^®^ (Madison, WI, USA) CellTiter-Glo^®^ (CTG) 2D cell viability assay.

### 3.12. Cell Viability Assay in 3D

HCT 116 cells were seeded in 96-well round bottom ultra-low attachment plates (1000 cells per well) and allowed to form spheroids for 72 h. All drugs were dissolved in DMSO, nanoparticles were suspended in DDW, and added to the wells. Untreated cells (control) were used to establish 100% viability. The effect on cell viability was measured with Promega^®^ CellTiter-Glo^®^ (CTG) 3D cell viability assay.

### 3.13. Drug Release Profile 

Ponatinib nanoparticles were incubated in serum-containing DMEM (pH 7.4) at 37 °C at a concentration equivalent to 50 μM of drug. The amount of released drug was determined by centrifugation at 3000 rcf for 10min followed by extraction of the drug from the supernatant into DMSO and fluorescence measurements at λ-excitation = 370 nm λ-emission = 450 nm using a plate reader (BioTeK SynergyN1). All experiments were carried out in duplicates.

## Figures and Tables

**Figure 1 ijms-23-12420-f001:**
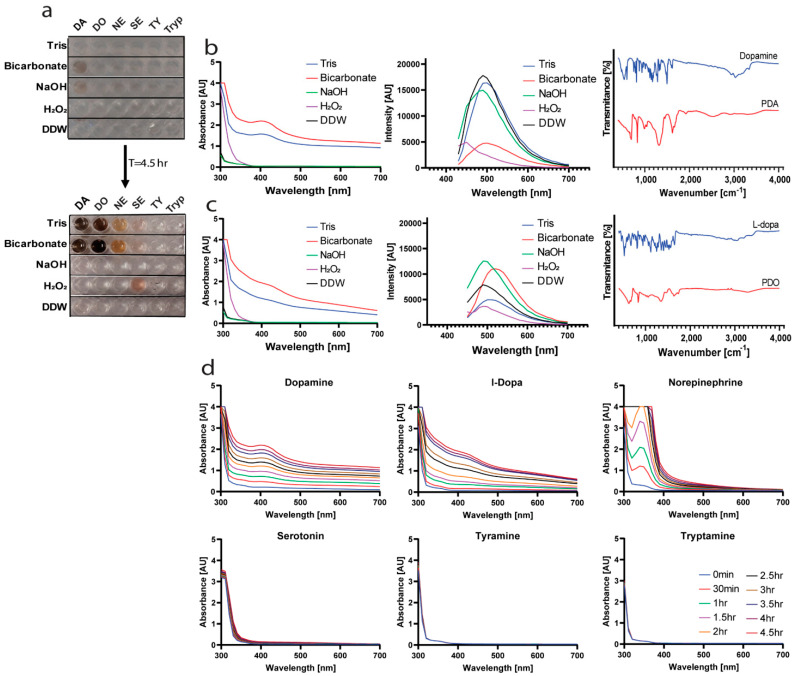
Polymerization process. (**a**) Different polymerization conditions and their impact on the polymerization of catecholamines after 4.5 h. Abbreviations: DA = dopamine, DO = L-dopa, NE = norepinephrine, SE = serotonin, TY = tyramine, Tryp = tryptamine. (**b**,**c**) Absorbance, emission, and ATR spectra of dopamine/PDA (**b**) and L-dopa/PDO (**c**). (**d**) Absorbance spectra of the reactions over time in bicarbonate buffer (0.1 M) and the reactant structures.

**Figure 2 ijms-23-12420-f002:**
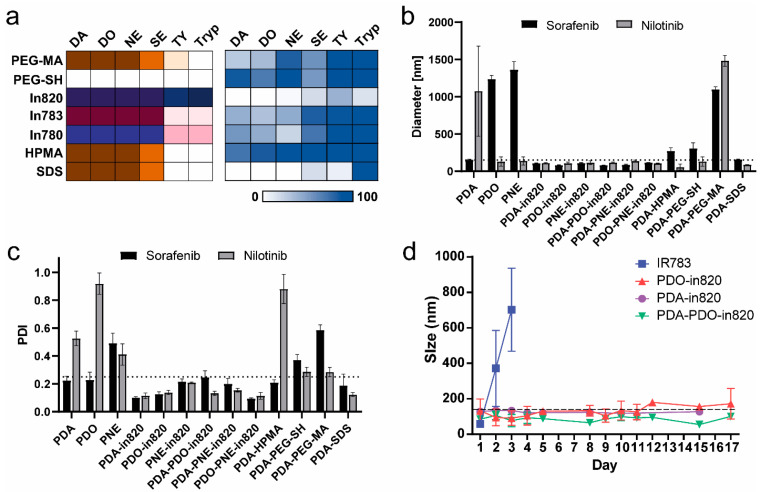
Selection of top copolymer. (**a**) Left comonomers matrix, illustration of colors after a 4 h reaction in sodium bicarbonate buffer 0.1 M. Right-normalized confluence of blue emission at 477 nm with nilotinib. (**b**) Average diameter of sorafenib and nilotinib nanoparticles formulated with different polymers. (**c**) Polydispersity index (PDI) of nanoparticles made with different copolymers. (**d**) Stabilization over time of nilotinib nanoparticles formulated with different stabilizers.

**Figure 3 ijms-23-12420-f003:**
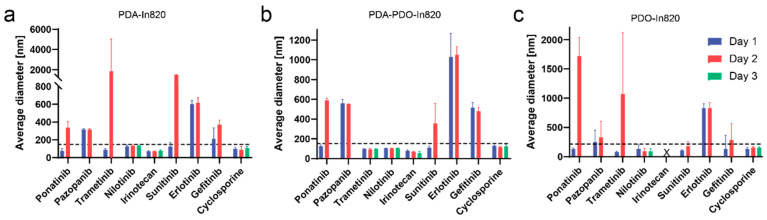
Three-day stability of 9 drugs stabilized by (**a**) PDA-In820, (**b**) PDA-PDO-In820, and (**c**) PDO-820.

**Figure 4 ijms-23-12420-f004:**

HR-SEM images of sorafenib nanoparticles formulated with (left to right) PDA-PDO-In820, PDA-In820, and PDO-In820, scale bars = 200 nm.

**Figure 5 ijms-23-12420-f005:**
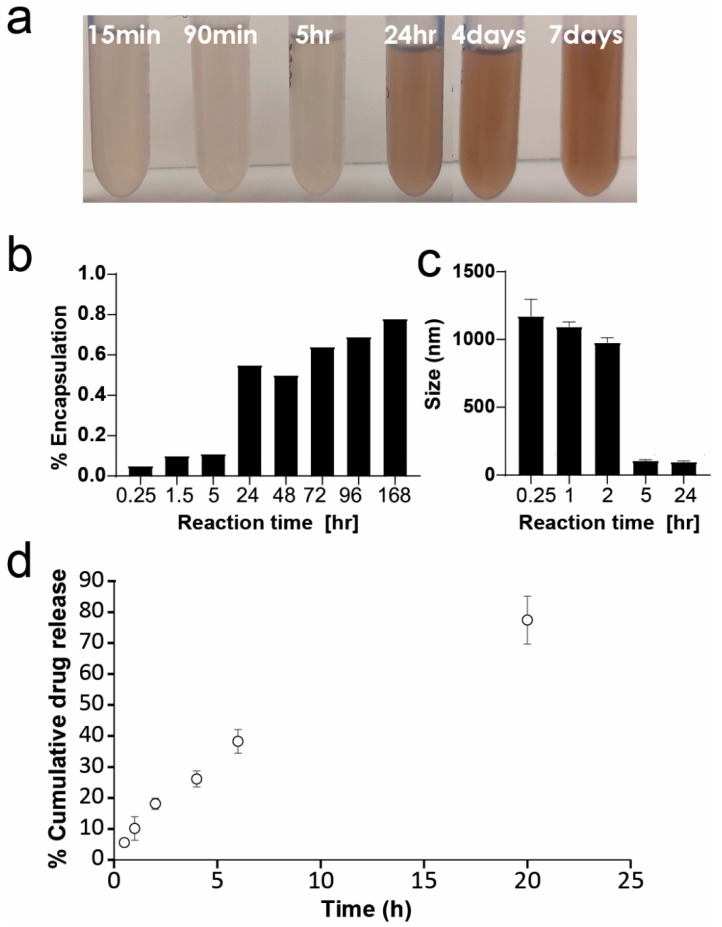
Characterization and optimization of PDA-PDO-In820. (**a**) Nilotinib nanoparticles formulated during different periods of time since polymerization initiation. (**b**) Encapsulation ratios of nilotinib nanoparticles from (**a**). (**c**) Size of ponatinib nanoparticles formulated in different time points during the aging process of PDA-PDO-In820. (**d**) Cumulative drug release profile of ponatinib nanoparticles using PDA-PDO-In820. Error bars represent standard deviation.

**Figure 6 ijms-23-12420-f006:**
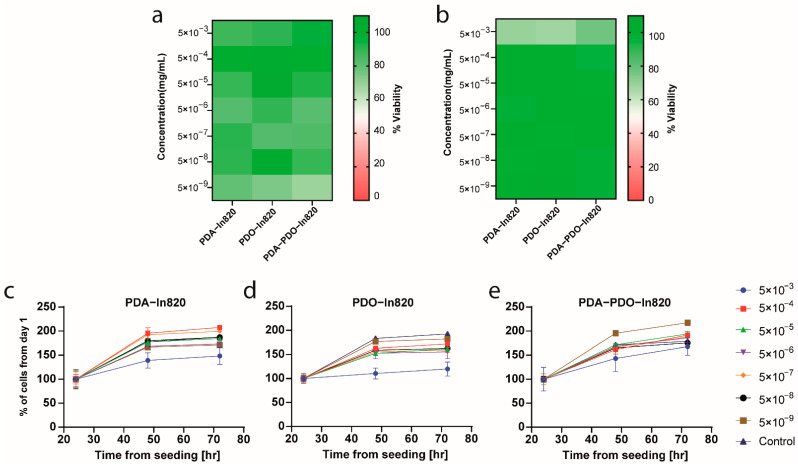
In vitro toxicity assay in mouse embryonic fibroblasts, 3T3 cells. (**a**) MTT after 3 days of incubation with different copolymers. (**b**) Cell viability assay using image-based cell count after 3 days of incubation with different copolymers. (**c**–**e**) Image-based cell counting at various times of incubation with different copolymers. Error bars represent standard deviation.

**Figure 7 ijms-23-12420-f007:**
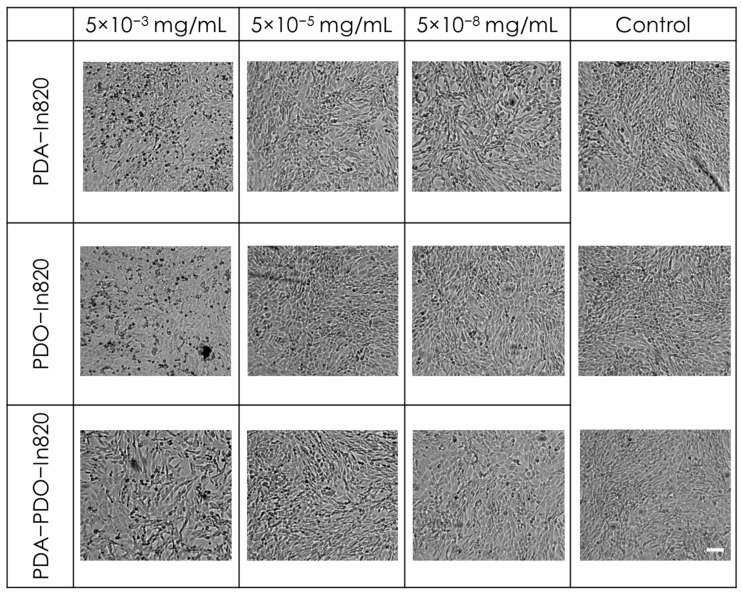
Images of 3T3 cell morphologies incubated with various concentrations of the different copolymers. Scale bar = 100 μm.

**Figure 8 ijms-23-12420-f008:**
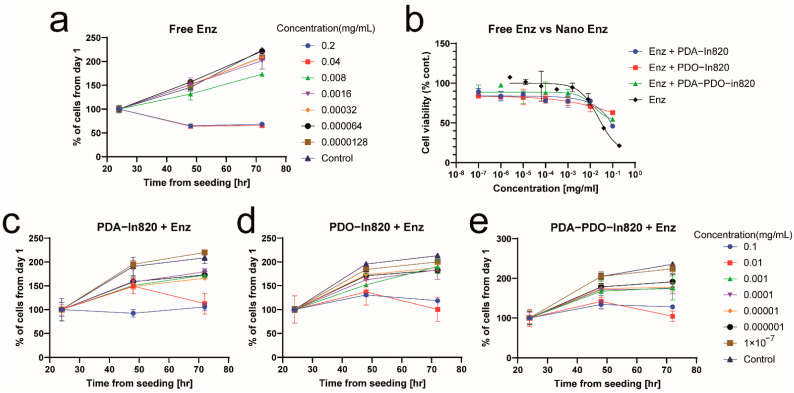
In vitro toxicity assay of enzalutamide nanoparticles in mouse embryonic fibroblasts, 3T3 cells. (**a**) Image-based cell counting at various times of incubation with free enzalutamide. (**b**) MTT viability assay after 3 days of incubation with free enzalutamide and its nanoparticles. (**c**–**e**) Image-based cell counting at various times of incubation with the different enzalutamide nanoparticles.

**Figure 9 ijms-23-12420-f009:**
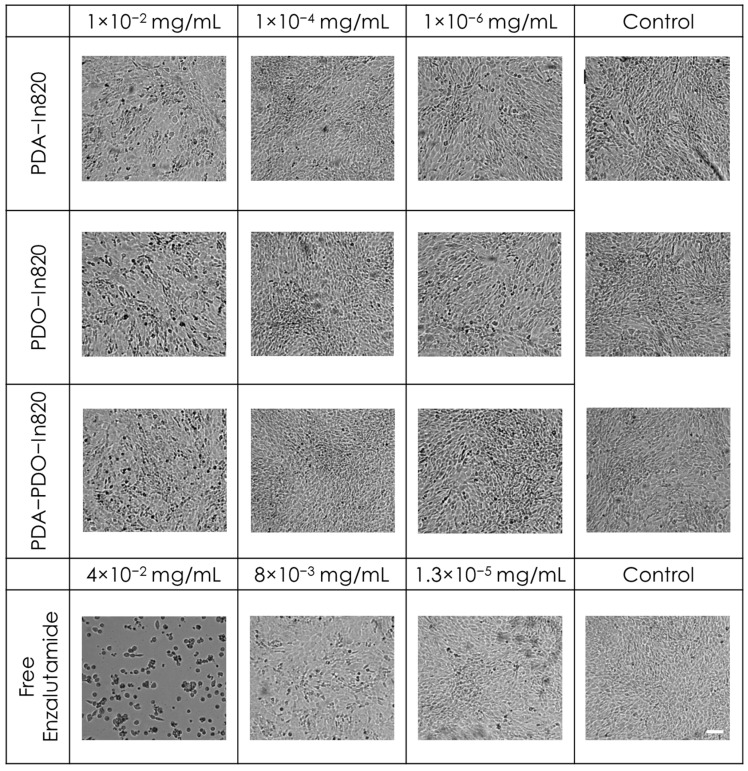
Images of 3T3 cell morphologies incubated with nanoparticles at various concentrations of the different copolymers and free enzalutamide. Scale bar = 100 μm.

**Figure 10 ijms-23-12420-f010:**
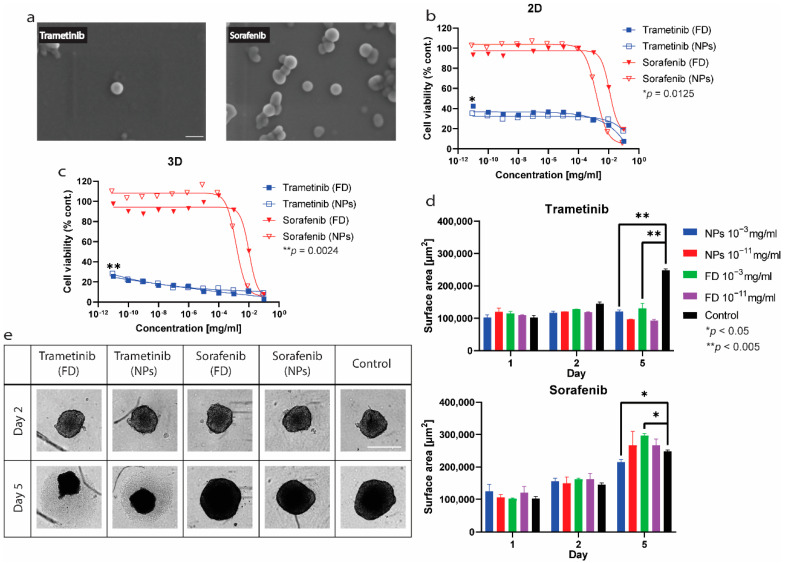
Cell viability assays in 2D and 3D models of HCT116. (**a**) HR-SEM images of the trametinib and sorafenib nanoparticles stabilized by PDA-PDO-In820, scale bar = 100 nm. (**b**) CTG 2D results of HCT 116 cells after 4 days of treatment. abbreviations: FD = free drug, NPs = nanoparticles. (**c**) CTG 3D results of HCT 116 spheroids after 5 days of treatment. (**d**) Change in spheroids size during the treatments. (**e**) Spheroids images on the 1st and 5th days of the treatment. Drugs and NPs concentrations = 10^−3^ mg/mL, scale bar = 500 μm.

## Data Availability

Data can be retrieved from corresponding author upon reasonable request.

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
