# Peer review of "Polydopamine Copolymers for Stable Drug Nanoprecipitation"

_ijms, 2022, doi:10.3390/ijms232012420_

Round 1

Reviewer 1 Report

Comment:

This manuscript is a very interesting study in which finding a polydopamine copolymers used for stable drug nanoprecipitation with high encapsulation efficiency. Several questions still need to be further answered or data supplemented.

1 the description about the preparation of drug nanoprecipitation is too rough. Please accurately state the added order of drugs and other components as well as other experimental details in Methods part.

2 in the part of 3.3, will changing the ratio of the three monomers affect the drug loading capacity?

3 what is the possible interaction between hydrophobic drugs and PDA-PDO-In820? Considering the DA and DO will polymerize in the alkaline solution, whether the drug and DA/DA can form covalent interaction? Which will affect the drug release behavior. As a drug delivery system, there lacks the data of drug release behavior in vitro. Please provide the drug release curve in vitro.

Author Response

Reviewer 1:

This manuscript is a very interesting study in which finding a polydopamine copolymers used for stable drug nanoprecipitation with high encapsulation efficiency. Several questions still need to be further answered or data supplemented.

  1. the description about the preparation of drug nanoprecipitation is too rough. Please accurately state the added order of drugs and other components as well as other experimental details in Methods
    1. The preparation of the nanoparticles was edited to a more chronological order, as well as other parts of the methods such as the polymerization process.
  2. in the part of 3.3, will changing the ratio of the three monomers affect the drug loading capacity?
    1. We have tested different ratios of the co-monomers, the resulted nanoparticles were not as reproducible as with the 1:1:1 ratio, moreover, their stabilizing abilities of some of them were inferior to it. Therefor we did not use them further or measured their encapsulation efficiency.
  3. what is the possible interaction between hydrophobic drugs and PDA-PDO-In820? Considering the DA and DO will polymerize in the alkaline solution, whether the drug and DA/DA can form covalent interaction? Which will affect the drug release behavior. As a drug delivery system, there lacks the data of drug release behavior in vitro. Please provide the drug release curve in vitro.
    1. Though it is indeed possible for a chemical reaction between DA/DO and the drug, we haven’t seen any evidence for this with the drugs that we tested. Using HPLC we found that the upon formulation drugs have the same retention time and thus are intact. The interactions are probably hydrophobic-pi-pi interactions as well as possible electrostatic interaction of the In820 and water. We do not think that the interactions between the DA or DO and the drug are covalent. We have performed in vitro drug release and observed a linear release profile which yielded 80% release in 20h. We provide the new data in figure 5d.

Reviewer 2 Report

 Manuscript – ijms-1940164

Polydopamine Copolymers for Stable Drug Nanoprecipitation

 ·  All abbreviations should be first identified before use them even if they were in Abstract or another part of the manuscript

·        There is an excessive use of acronyms, which makes the reading sometimes difficult

·       The abstract does not report the main findings of the study in a clear manner. For example, general expressions are used which do not provide useful information to the readers. Information that is more specific is required in the abstract.

·       The introduction does not point out the gap of the literature the study seeks to fill and novelty of the study over the existing literature. This point showed be further elaborated.   

·       A relevant hypothesis for the study is missing from the introduction. A true scientific question should be formed

·       Simplify the statement in the paper. Please combine and condense the discussion and conclusion

·       Enhance the resolution of figures 2 (b and c) and 9

·       doi should always be added when available

Author Response

Reviewer 2:

  1. All abbreviations should be first identified before use them even if they were in Abstract or another part of the manuscript
    1. Abbreviations that were not specified are now cleared.
  2. There is an excessive use of acronyms, which makes the reading sometimes difficult
    1. The use of acronyms was reduced to only when necessary.
  3. The abstract does not report the main findings of the study in a clear manner. For example, general expressions are used which do not provide useful information to the readers. Information that is more specific is required in the abstract.
    1. Some key results regarding PDA-PDO-In820 ability to formulate stable and improved nanoparticles were added to the abstract to be highlighted.
  4. The introduction does not point out the gap of the literature the study seeks to fill and novelty of the study over the existing literature. This point showed be further elaborated.   
    1. We added a more focused paragraph on this issue. The gap in the literature we seek to fill is the incorporation of polydopamine and poly(catecholamine)s as stabilizers into drug nanoprecipitation. We were inspired by the use of polydopamine in biomedicine and drug delivery and want to utilize it in a different method that is commonly used for drug nanoparticles formation but is challenging because these materials tend to adhere and aggregate.
  5. A relevant hypothesis for the study is missing from the introduction. A true scientific question should be formed
    1. The hypothesis was added to the introduction. We hypothesis that dopamine-based stabilizers can have a significant contribution to precipitated nanoparticles and improve the formulations of different drugs.
  6. Simplify the statement in the paper. Please combine and condense the discussion and conclusion
    1. The conclusions and discussion sections were combined and only the key results are stated as the conclusions.
  7. Enhance the resolution of figures 2 (b and c) and 9
    1. The resolution and overall appearance of these figures were improved.
  8. doi should always be added when available
    1. DOI was added to all references

Reviewer 3 Report

Great paper!

Author Response

We thank the reviewer for the positive comments.

Round 2

Reviewer 1 Report

The author has clearly answered my questions and I suggested that the article can be accepted.

Reviewer 2 Report

Dear Editor Journal of international journal of molecular science

Manuscript ID: ID: ijms-1940164

I re-reviewed the manuscript “Polydopamine Copolymers for Stable Drug Nanoprecipitationagain and the authors made all the amendments that I asked before so I think the manuscript is suitable for publishing

Regards